# *LEADS*: Learning Dynamical Systems that Generalize Across Environments

**Yuan Yin[1], Ibrahim Ayed[1,2], Emmanuel de Bézenac[1], Nicolas Baskiotis[1], Patrick Gallinari[1,3]**
[1]Sorbonne Université, Paris, France
[2]ThereSIS Lab, Thales, Paris, France    [3]Criteo AI Lab, Paris, France
{yuan.yin,ibrahim.ayed,emmanuel.de-bezenac,
nicolas.baskiotis,patrick.gallinari}@sorbonne-universite.fr

## Abstract

When modeling dynamical systems from real-world data samples, the distribution of data often changes according to the environment in which they are captured, and the dynamics of the system itself vary from one environment to another. Generalizing across environments thus challenges the conventional frameworks. The classical settings suggest either considering data as i.i.d. and learning a single model to cover all situations or learning environment-specific models. Both are suboptimal: the former disregards the discrepancies between environments leading to biased solutions, while the latter does not exploit their potential commonalities and is prone to scarcity problems. We propose *LEADS*, a novel framework that leverages the commonalities and discrepancies among known environments to improve model generalization. This is achieved with a tailored training formulation aiming at capturing common dynamics within a shared model while additional terms capture environment-specific dynamics. We ground our approach in theory, exhibiting a decrease in sample complexity w.r.t. classical alternatives. We show how theory and practice coincides on the simplified case of linear dynamics. Moreover, we instantiate this framework for neural networks and evaluate it experimentally on representative families of nonlinear dynamics. We show that this new setting can exploit knowledge extracted from environment-dependent data and improves generalization for both known and novel environments.

## 1   Introduction

Data-driven approaches offer an interesting alternative and complement to physical-based methods for modeling the dynamics of complex systems and are particularly promising in a wide range of settings: e.g. if the underlying dynamics are partially known or understood, if the physical model is incomplete, inaccurate, or fails to adapt to different contexts, or if external perturbation sources and forces are not modeled. The idea of deploying machine learning (ML) to model complex dynamical systems picked momentum a few years ago, relying on recent deep learning progresses and on the development of new methods targeting the evolution of temporal and spatiotemporal systems [6, 9, 7, 21, 30, 2, 37]. It is already being applied in different scientific disciplines (see e.g. [36] for a recent survey) and could help accelerate scientific discovery to address challenging domains such as climate [32] or health [12].

However, despite promising results, current developments are limited and usually postulate an idealized setting where data is *abundant* and *the environment does not change*, the so-called "i.i.d. hypothesis". In practice, real-world data may be expensive or difficult to acquire. Moreover, changes in the environment may be caused by many different factors. For example, in climate modeling, there are external forces (e.g. Coriolis) which depend on the spatial location [23]; or, in health science, parameters need to be personalized for each patient as for cardiac computational models [27]. More

generally, data acquisition and modeling are affected by different factors such as geographical position, sensor variability, measuring circumstances, etc. The classical paradigm either considers all the data as i.i.d. and looks for a global model, or proposes specific models for each environment. The former disregards discrepancies between the environments, thus leading to a biased solution with an averaged model which will usually perform poorly. The latter ignores the similarities between environments, thus affecting generalization performance, particularly in settings where per-environment data is limited. This is particularly problematic in dynamical settings, as small changes in initial conditions lead to trajectories not covered by the training data.

In this work, we consider a setting where it is explicitly assumed that the trajectories are collected from different environments. Note that in this setting, the i.i.d. hypothesis is removed twice: by considering the temporality of the data and by the existence of multiple environments. In many useful contexts the dynamics in each environment share similarities, while being distinct which translates into changes in the data distributions. Our objective is to leverage the similarities between environments in order to improve the modeling capacity and generalization performance, while still carefully dealing with the discrepancies across environments. This brings us to consider two research questions:

**RQ1** Does modeling the differences between environments improve generalization error w.r.t. classical settings: ***One-For-All***, where a unique function is trained for all environments; and ***One-Per-Env.***, where a specific function is fitted for each environment? (cf. Sec. 4 for more details)

**RQ2** Is it possible to extrapolate to a novel environment that has not been seen during training?

We propose LEarning Across Dynamical Systems (*LEADS*), a novel learning methodology decomposing the learned dynamics into *shared* and *environment-specific* components. The learning problem is formulated such that the *shared* component captures the dynamics common across environments and exploits all the available data, while the *environment-specific* component only models the remaining dynamics, i.e. those that cannot be expressed by the former, based on environment-specific data. We show, under mild conditions, that the learning problem is well-posed, as the resulting decomposition exists and is unique (Sec. 2.2). We then analyze the properties of this decomposition from a sample complexity perspective. While, in general, the bounds might be too loose to be practical, a more precise study is conducted in the case of linear dynamics for which theory and practice are closer. We then instantiate this framework for more general hypothesis spaces and dynamics, leading to a heuristic for the control of generalization that will be validated experimentally. Overall, we show that this framework provides better generalization properties than *One-Per-Env.*, requiring less training data to reach the same performance level (*RQ1*). The shared information is also useful to extrapolate to unknown environments: the new function for this environment can be learned from very little data (*RQ2*). We experiment with these ideas on three representative cases (Sec. 4) where the dynamics are provided by differential equations: ODEs with the Lotka-Volterra predator-prey model, and PDEs with the Gray-Scott reaction-diffusion and the more challenging incompressible Navier-Stokes equations. Experimental evidence confirms the intuition and the theoretical findings: with a similar amount of data, the approach drastically outperforms *One-For-All* and *One-Per-Env.* settings, especially in low data regimes. Up to our knowledge, it is the first time that generalization in multiple dynamical systems is addressed from an ML perspective[1].

## 2 Approach

### 2.1 Problem setting

We consider the problem of learning models of dynamical physical processes with data acquired from a set of environments $E$. Throughout the paper, we will assume that the dynamics in an environment $e \in E$ are defined through the evolution of differential equations. This will provide in particular a clear setup for the experiments and the validation. For a given problem, we consider that the dynamics of the different environments share common factors while each environments has its own specificity, resulting in a distinct model per environment. Both the general form of the differential equations and the specific terms of each environment are assumed to be completely unknown. $x_t^e$ denotes the state of the equation for environment $e$, taking its values from a bounded set $\mathcal{A}$, with evolution term $f_e : \mathcal{A} \rightarrow T\mathcal{A}$, $T\mathcal{A}$ being the tangent bundle of $\mathcal{A}$. In other words, over a fixed time interval $[0, T]$, we have:

$$\frac{\mathrm{d}x_t^e}{\mathrm{d}t} = f_e(x_t^e) \tag{1}$$

---

[1] Code is available at https://github.com/yuan-yin/LEADS.

We assume that, for any $e$, $f_e$ lies in a functional vector space $\mathcal{F}$. In the experiments, we will consider one ODE, in which case $\mathcal{A} \subset \mathbb{R}^d$, and two PDEs, in which case $\mathcal{A}$ is a $d'$-dimensional vector field over a bounded spatial domain $S \subset \mathbb{R}^{d'}$. The term of the data-generating dynamical system in Eq. 1 is sampled from a distribution for each $e$, i.e. $f_e \sim Q$. From $f_e$, we define $\mathcal{T}_e$, the data distribution of trajectories $x^e$ verifying Eq. 1, induced by a distribution of initial states $x_0^e \sim P_0$. The data for this environment is then composed of $l$ trajectories sampled from $\mathcal{T}_e$, and is denoted as $\hat{\mathcal{T}}_e$ with $x^{e,i}$ the $i$-th trajectory. We will denote the full dataset by $\hat{\mathcal{T}} = \bigcup_{e \in E} \hat{\mathcal{T}}_e$.

The classical empirical risk minimization (ERM) framework suggests to model the data dynamics either at the global level (*One-For-All*), taking trajectories indiscriminately from $\hat{\mathcal{T}}$, or at the specific environment level (*One-Per-Env.*), training one model for each $\hat{\mathcal{T}}_e$. Our aim is to formulate a new learning framework with the objective of explicitly considering the existence of different environments to improve the modeling strategy w.r.t. the classical ERM settings.

## 2.2 *LEADS* framework

We decompose the dynamics into two components where $f \in \mathcal{F}$ is shared across environments and $g_e \in \mathcal{F}$ is specific to the environment $e$, so that

$$\forall e \in E, f_e = f + g_e \tag{2}$$

Since we consider functional vector spaces, this additive hypothesis is not restrictive and such a decomposition always exists. It is also quite natural as a sum of evolution terms can be seen as the sum of the forces acting on the system. Note that the sum of two evolution terms can lead to behaviors very different from those induced by each of those terms. However, learning this decomposition from data defines an ill-posed problem: for any choice of $f$, there is a $\{g_e\}_{e \in E}$ such that Eq. 2 is verified. A trivial example would be $f = 0$ leading to a solution where each environment is fitted separately.

Our core idea is that $f$ should capture as much of the shared dynamics as is possible, while $g_e$ should focus only on the environment characteristics not captured by $f$. To formalize this intuition, we introduce $\Omega(g_e)$, a penalization on $g_e$, which precise definition will depend on the considered setting. We reformulate the learning objective as the following constrained optimization problem:

$$\min_{f, \{g_e\}_{e \in E} \in \mathcal{F}} \sum_{e \in E} \Omega(g_e) \quad \text{subject to} \quad \forall x^{e,i} \in \hat{\mathcal{T}}, \forall t, \frac{\mathrm{d}x_t^{e,i}}{\mathrm{d}t} = (f + g_e)(x_t^{e,i}) \tag{3}$$

Minimizing $\Omega$ aims to reduce $g_e$s' complexity while correctly fitting the dynamics of each environment. This argument will be made formal in the next section. Note that $f$ will be trained on the data from all environments contrary to $g_e$s. A key question is then to determine under which conditions the minimum in Eq. 3 is well-defined. The following proposition provides an answer (proof cf. Sup. A):

**Proposition 1** (Existence and Uniqueness). *Assume $\Omega$ is convex, then the existence of a minimal decomposition $f^\star, \{g_e^\star\}_{e \in E} \in \mathcal{F}$ of Eq. 3 is guaranteed. Furthermore, if $\Omega$ is strictly convex, this decomposition is unique.*

In practice, we consider the following relaxed formulation of Eq. 3:

$$\min_{f, \{g_e\}_{e \in E} \in \hat{\mathcal{F}}} \sum_{e \in E} \left( \frac{1}{\lambda} \Omega(g_e) + \sum_{i=1}^{l} \int_0^T \left\| \frac{\mathrm{d}x_t^{e,i}}{\mathrm{d}t} - (f + g_e)(x_\tau^{e,i}) \right\|^2 \mathrm{d}t \right) \tag{4}$$

where $f, g_e$ are taken from a hypothesis space $\hat{\mathcal{F}}$ approximating $\mathcal{F}$. $\lambda$ is a regularization weight and the integral term constrains the learned $f + g_e$ to follow the observed dynamics. The form of this objective and its effective calculation will be detailed in Sec. 4.4.

## 3 Improving generalization with *LEADS*

Defining an appropriate $\Omega$ is crucial for our method. In this section, we show that the generalization error should decrease with the number of environments. While the bounds might be too loose for NNs, our analysis is shown to adequately model the decreasing trend in the linear case, linking both our intuition and our theoretical analysis with empirical evidence. This then allows us to construct an appropriate $\Omega$ for NNs.

### 3.1 General case

After introducing preliminary notations and definitions, we define the hypothesis spaces associated with our multiple environment framework. Considering a first setting where all environments of interest are present at training time, we prove an upper-bound of their effective size based on the

covering numbers of the approximation spaces. This allows us to quantitatively control the sample complexity of our model, depending on the number of environments $m$ and other quantities that can be considered and optimized in practice. We then consider an extension for learning on a new and unseen environment. The bounds here are inspired by ideas initially introduced in [4]. They consider multi-task classification in vector spaces, where the task specific classifiers share a common feature extractor. Our extension considers sequences corresponding to dynamical trajectories, and a model with additive components instead of function composition in their case.

**Definitions.** Sample complexity theory is usually defined for supervised contexts, where for a given input $x$ we want to predict some target $y$. In our setting, we want to learn trajectories $(x_t^e)_{0 \le t \le T}$ starting from an initial condition $x_0$. We reformulate this problem and cast it as a standard supervised learning problem: $\mathcal{T}_e$ being the data distribution of trajectories for environment $e$, as defined in Sec. 2.1, let us consider a trajectory $x_\cdot^e \sim \mathcal{T}_e$, and time $\tau \sim \mathrm{Unif}([0,T])$; we define system states $x = x_\tau^e \in \mathcal{A}$ as input, and the corresponding values of derivatives $y = f_e(x_\tau^e) \in T\mathcal{A}$ as the associated target. We will denote $\mathcal{P}_e$ the underlying distribution of $(x, y)$, and $\hat{\mathcal{P}}_e$ the associated dataset of size $n$.

We are searching for $f, g_e : \mathcal{A} \to T\mathcal{A}$ in an approximation function space $\hat{\mathcal{F}}$ of the original space $\mathcal{F}$. Let us define $\hat{\mathcal{G}} \subseteq \hat{\mathcal{F}}$ the effective function space from which the $g_e$s are sampled. Let $f + \hat{\mathcal{G}} := \{f + g : g \in \hat{\mathcal{G}}\}$ be the hypothesis space generated by function pairs $(f, g)$, with a fixed $f \in \hat{\mathcal{F}}$. For any $h : \mathcal{A} \to T\mathcal{A}$, the error on some test distribution $\mathcal{P}_e$ is given by $\mathrm{er}_{\mathcal{P}_e}(h) = \int_{\mathcal{A} \times T\mathcal{A}} \|h(x) - y\|^2 \mathrm{d}\mathcal{P}_e(x, y)$ and the error on the training set by $\hat{\mathrm{er}}_{\hat{\mathcal{P}}_e}(h) = \frac{1}{n} \sum_{(x,y) \in \hat{\mathcal{P}}_e} \|h(x) - y\|^2$.

***LEADS* sample complexity.** Let $\mathcal{C}_{\hat{\mathcal{G}}}(\varepsilon, \hat{\mathcal{F}})$ and $\mathcal{C}_{\hat{\mathcal{F}}}(\varepsilon, \hat{\mathcal{G}})$ denote the capacity of $\hat{\mathcal{F}}$ and $\hat{\mathcal{G}}$ at a certain scale $\varepsilon > 0$. Such capacity describes the approximation ability of the space. The capacity of a class of functions is defined based on covering numbers, and the precise definition is provided in Sup. B.2, Table S1. The following result is general and applies for *any* decomposition of the form $f + g_e$. It states that to guarantee a given average test error, the minimal number of samples required is a function of both capacities and the number of environments $m$, and it provides a step towards *RQ1* (proof see Sup. B.2):

**Proposition 2.** *Given $m$ environments, let $\varepsilon_1, \varepsilon_2, \delta > 0, \varepsilon = \varepsilon_1 + \varepsilon_2$. Assume the number of examples $n$ per environment satisfies*

$$n \ge \max \left\{ \frac{64}{\varepsilon^2} \left( \frac{1}{m} \left( \log \frac{4}{\delta} + \log \mathcal{C}_{\hat{\mathcal{G}}}\left(\frac{\varepsilon_1}{16}, \hat{\mathcal{F}}\right) \right) + \log \mathcal{C}_{\hat{\mathcal{F}}}\left(\frac{\varepsilon_2}{16}, \hat{\mathcal{G}}\right) \right), \frac{16}{\varepsilon^2} \right\} \tag{5}$$

*Then with probability at least $1 - \delta$ (over the choice of training sets $\{\hat{\mathcal{P}}_e\}$), any learner $(f + g_1, \ldots, f + g_m)$ will satisfy $\frac{1}{m} \sum_{e \in E} \mathrm{er}_{\mathcal{P}_e}(f + g_e) \le \frac{1}{m} \sum_{e \in E} \hat{\mathrm{er}}_{\hat{\mathcal{P}}_e}(f + g_e) + \varepsilon$.*

The contribution of $\hat{\mathcal{F}}$ to the sample complexity decreases as $m$ increases, while that of $\hat{\mathcal{G}}$ remains the same: this is due to the fact that shared functions $f$ have access to the data from all environments, which is not the case for $g_e$. From this finding, one infers the basis of *LEADS*: when learning from several environments, to control the generalization error through the decomposition $f_e = f + g_e$, $f$ *should account for most of the complexity of $f_e$ while the complexity of $g_e$ should be controlled and minimized*. We then establish an explicit link to our learning problem formulation in Eq. 3. Further in this section, we will show for linear ODEs that the optimization of $\Omega(g_e)$ in Eq. 4 controls the capacity of the effective set $\hat{\mathcal{G}}$ by selecting $g_e$s that are as "simple" as possible.

As a corollary, we show that for a fixed total number of samples in $\hat{\mathcal{T}}$, the sample complexity will decrease as the number of environments increases. To see this, suppose that we have two situations corresponding to data generated respectively from $m$ and $m/b$ environments. The total sample complexity for each case will be respectively bounded by $O(\log \mathcal{C}_{\hat{\mathcal{G}}}(\frac{\varepsilon_1}{16}, \hat{\mathcal{F}}) + m \log \mathcal{C}_{\hat{\mathcal{F}}}(\frac{\varepsilon_2}{16}, \hat{\mathcal{G}}))$ and $O(b \log \mathcal{C}_{\hat{\mathcal{G}}}(\frac{\varepsilon_1}{16}, \hat{\mathcal{F}}) + m \log \mathcal{C}_{\hat{\mathcal{F}}}(\frac{\varepsilon_2}{16}, \hat{\mathcal{G}}))$. The latter being larger than the former, a situation with more environments presents a clear advantage. Fig. 4 in Sec. 4 confirms this result with empirical evidence.

***LEADS* sample complexity for novel environments.** Suppose that problem Eq. 3 has been solved for a set of environments $E$, can we use the learned model for a new environment not present in the initial training set (*RQ2*)? Let $e'$ be such a new environment, $\mathcal{P}_{e'}$ the trajectory distribution of $e'$, generated from dynamics $f_{e'} \sim Q$, and $\hat{\mathcal{P}}_{e'}$ an associated training set of size $n'$. The following results show that the number of required examples for reaching a given performance is much lower when training $f + g_{e'}$ with $f$ fixed on this new environment than training another $f' + g_{e'}$ from scratch (proof see Sup. B.2).

**Proposition 3.** *For all $\varepsilon, \delta$ with $0 < \varepsilon, \delta < 1$ if the number of samples $n'$ satisfies*

$$n' \geq \max\left\{ \frac{64}{\varepsilon^2} \log \frac{4\mathcal{C}(\frac{\varepsilon}{16}, f + \hat{\mathcal{G}})}{\delta}, \frac{16}{\varepsilon^2} \right\}, \tag{6}$$

*then with probability at least $1 - \delta$ (over the choice of novel training set $\hat{\mathcal{P}}_{e'}$), any learner $f + g_{e'} \in f + \hat{\mathcal{G}}$ will satisfy $\mathrm{er}_{\mathcal{P}_{e'}}(f + g_{e'}) \leq \hat{\mathrm{er}}_{\hat{\mathcal{P}}_{e'}}(f + g_{e'}) + \varepsilon$.*

In Prop. 3 as the capacity of $\hat{\mathcal{F}}$ no longer appears, the number of required samples now depends only on the capacity of $f + \hat{\mathcal{G}}$. This sample complexity is then smaller than learning from scratch $f_{e'} = f + g_{e'}$ as can be seen by comparing with Prop. 2 at $m = 1$.

From the previous propositions, it is clear that the environment-specific functions $g_e$ need to be explicitly controlled. We now introduce a practical way to do that. Let $\omega(r, \varepsilon)$ be a strictly increasing function w.r.t. $r$ such that

$$\log \mathcal{C}_{\hat{\mathcal{F}}}(\varepsilon, \hat{\mathcal{G}}) \leq \omega(r, \varepsilon), \quad r = \sup_{g \in \hat{\mathcal{G}}} \Omega(g) \tag{7}$$

Minimizing $\Omega$ would reduce $r$ and then the sample complexity of our model by constraining $\hat{\mathcal{G}}$. Our goal is thus to construct such a pair $(\omega, \Omega)$. In the following, we will first show in Sec. 3.2, how one can construct a penalization term $\Omega$ based on the covering number bound for linear approximators and linear ODEs. We show with a simple use case that the generalization error obtained in practice follows the same trend as the theoretical error bound when the number of environments varies. Inspired by this result, we then propose in Sec. 3.3 an effective $\Omega$ to penalize the complexity of the neural networks $g_e$.

### 3.2 Linear case: theoretical bounds correctly predict the trend of test error

Results in Sec. 3.1 provide general guidelines for our approach. We now apply them to a linear system to see how the empirical results meet the tendency predicted by theoretical bound.

Let us consider a linear ODE $\frac{\mathrm{d}x_t^e}{\mathrm{d}t} = \mathrm{L}_{\boldsymbol{F}_e}(x_t^e)$ where $\mathrm{L}_{\boldsymbol{F}_e} : x \mapsto \boldsymbol{F}_e x$ is a linear transformation associated to the square real valued matrix $\boldsymbol{F}_e \in M_{d,d}(\mathbb{R})$. We choose as hypothesis space the space of linear functions $\hat{\mathcal{F}} \subset \mathcal{L}(\mathbb{R}^d, \mathbb{R}^d)$ and instantiate a linear *LEADS* $\frac{\mathrm{d}x_t^e}{\mathrm{d}t} = (\mathrm{L}_{\boldsymbol{F}} + \mathrm{L}_{\boldsymbol{G}_e})(x_t^e)$, $\mathrm{L}_{\boldsymbol{F}} \in \hat{\mathcal{F}}, \mathrm{L}_{\boldsymbol{G}_e} \in \hat{\mathcal{G}} \subseteq \hat{\mathcal{F}}$. As suggested in [3], we have that (proof in Sup. B.3):

**Proposition 4.** *If for all linear maps $\mathrm{L}_{\boldsymbol{G}_e} \in \hat{\mathcal{G}}$, $\|\boldsymbol{G}\|_F^2 \leq r$, if the input space is bounded s.t. $\|x\|_2 \leq b$, and the MSE loss function is bounded by $c$, then*

$$\log \mathcal{C}_{\hat{\mathcal{F}}}(\varepsilon, \hat{\mathcal{G}}) \leq \lceil rcd(2b)^2/\varepsilon^2 \rceil \log 2d^2 =: \omega(r, \varepsilon)$$

$\omega(r, \varepsilon)$ is a strictly increasing function w.r.t. $r$. This indicates that we can choose $\Omega(\mathrm{L}_{\boldsymbol{G}}) = \|\boldsymbol{G}\|_F$ as our optimization objective in Eq. 3. The sample complexity in Eq. 5 will decrease with the size the largest possible $r = \sup_{\mathrm{L}_{\boldsymbol{G}} \in \hat{\mathcal{G}}} \Omega(\mathrm{L}_{\boldsymbol{G}})$. The optimization process will reduce $\Omega(\mathrm{L}_{\boldsymbol{G}})$ until a minimum is reached. The maximum size of the effective hypothesis space is then bounded and decreases throughout training thanks to the penalty. Then in linear case Prop. 2 becomes (proof cf. Sup. B.3):

**Proposition 5.** *If for linear maps $\mathrm{L}_{\boldsymbol{F}} \in \hat{\mathcal{F}}$, $\|\boldsymbol{F}\|_F^2 \leq r'$, $\mathrm{L}_{\boldsymbol{G}} \in \hat{\mathcal{G}}$, $\|\boldsymbol{G}\|_F^2 \leq r$, $\|x\|_2 \leq b$, and if the MSE loss function is bounded by $c$, given $m$ environments and $n$ samples per environment, with the probability $1 - \delta$, the generalization error upper bound is $\varepsilon = \max\left\{ \sqrt{(p + \sqrt{p^2 + 4q})/2}, \sqrt{16/n} \right\}$ where $p = \frac{64}{mn} \log \frac{4}{\delta}$ and $q = \frac{64}{n} \lceil (\frac{r'}{mz^2} + \frac{r}{(1-z)^2}) cd(32b)^2 \rceil \log 2d^2$ for any $0 < z < 1$.*

In Fig. 1, we take an instance of linear ODE defined by $\boldsymbol{F}_e = \boldsymbol{Q}\boldsymbol{\Lambda}_e\boldsymbol{Q}^\top$ with the diagonal $\boldsymbol{\Lambda}_e$ specific to each environment After solving Eq. 3 we have at the optimum that $\boldsymbol{G}_e = \boldsymbol{F}_e - \boldsymbol{F}^\star = \boldsymbol{F}_e - \frac{1}{m}\sum_{e' \in E} \boldsymbol{F}_{e'}$. Then we can take $r = \max_{\{\mathrm{L}_{\boldsymbol{G}_e}\}} \Omega(\mathrm{L}_{\boldsymbol{G}_e})$ as the norm bound of $\hat{\mathcal{G}}$ when $\Omega(g_e)$ is optimized. Fig. 1 shows on the left the test error with and without penalty and the corresponding theoretical bound on the right. We observe that, after applying the penalty $\Omega$, the test error is reduced as well as the theoretical generalization bound, as indicated by the arrows from the dashed line to the concrete one. See Sup. B.3 for more details on this experiment.

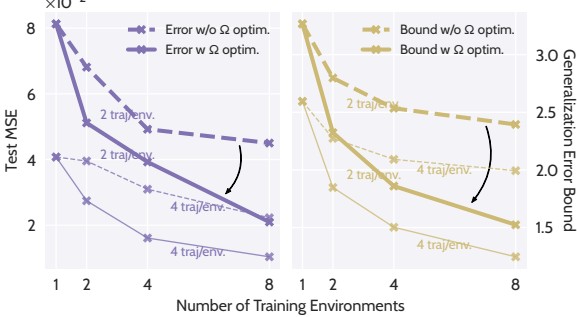

Figure 1: Test error compared with corresponding theoretical bound. The arrows indicate the changes after applying $\Omega(g_e)$ penalty.

## 3.3 Nonlinear case: instantiation for neural nets

The above linear case validates the ideas introduced in Prop. 2 and provides an instantiation guide and an intuition on the more complex nonlinear case. This motivates us to instantiate the general case by choosing an appropriate approximating space $\hat{\mathcal{F}}$ and a penalization function $\Omega$ from the generalization bounds for the corresponding space. Sup. B.4 of the Appendix contains additional details justifying those choices. For $\hat{\mathcal{F}}$, we select the space of feed-forward neural networks with a fixed architecture. We choose the following penalty function:

$$\Omega(g_e) = \|g_e\|_\infty^2 + \alpha \|g_e\|_{\text{Lip}}^2 \tag{8}$$

where $\|g\|_\infty = \operatorname{ess\,sup}|g|$ and $\|\cdot\|_{\text{Lip}}$ is the Lipschitz semi-norm, $\alpha$ is a hyperparameter. This is inspired by the existing capacity bound for NNs [14] (see Sup. B.4 for details). Note that constructing tight generalization bounds for neural networks is still an open research problem [26]; however, it may still yield valuable intuitions and guide algorithm design. This heuristic is tested successfully on three different datasets with different architectures in the experiments (Sec. 4).

# 4 Experiments

Our experiments are conducted on three families of dynamical systems described by three broad classes of differential equations. All exhibit complex and nonlinear dynamics. The first one is an ODE-driven system used for biological system modeling. The second one is a PDE-driven reaction-diffusion model, well-known in chemistry for its variety of spatiotemporal patterns. The third one is the more physically complex Navier-Stokes equation, expressing the physical laws of incompressible Newtonian fluids. To show the general validity of our framework, we will use 3 different NN architectures (MLP, ConvNet, and Fourier Neural Operator [19]). Each architecture is well-adapted to the corresponding dynamics. This also shows that the framework is valid for a variety of approximating functions.

## 4.1 Dynamics, environments, and datasets

**Lotka-Volterra (LV).**    This classical model [22] is used for describing the dynamics of interaction between a predator and a prey. The dynamics follow the ODE:

$$\mathrm{d}u/\mathrm{d}t = \alpha u - \beta uv, \, \mathrm{d}v/\mathrm{d}t = \delta uv - \gamma v$$

with $u, v$ the number of prey and predator, $\alpha, \beta, \gamma, \delta > 0$ defining how the two species interact. The system state is $x_t^e = (u_t^e, v_t^e) \in \mathbb{R}_+^2$. The initial conditions $u_0^i, v_0^i$ are sampled from a uniform distribution $P_0$. We characterize the dynamics by $\theta = (\alpha/\beta, \gamma/\delta) \in \Theta$. An environment $e$ is then defined by parameters $\theta_e$ sampled from a uniform distribution over a parameter set $\Theta$. We then sample two sets of environment parameters: one used as training environments for *RQ1*, the other treated as novel environments. for *RQ2*.

**Gray-Scott (GS).**    This reaction-diffusion model is famous for its complex spatiotemporal behavior given its simple equation formulation [29]. The governing PDE is:

$$\partial u/\partial t = D_u \Delta u - uv^2 + F(1-u), \, \partial v/\partial t = D_v \Delta v + uv^2 - (F+k)v$$

where the $u, v$ represent the concentrations of two chemical components in the spatial domain $S$ with periodic boundary conditions, the spatially discretized state at time $t$ is $x_t^e = (u_t^e, v_t^e) \in \mathbb{R}_+^{2 \times 32^2}$. $D_u, D_v$ denote the diffusion coefficients respectively for $u, v$, and are held constant, and $F, k$ are the reaction parameters determining the spatio-temporal patterns of the dynamics [29]. As for the initial conditions $(u_0, v_0) \sim P_0$, we consider uniform concentrations, with 3 2-by-2 squares fixed at other concentration values and positioned at uniformly sampled positions in $S$ to trigger the reactions. An environment $e$ is defined by its parameters $\theta_e = (F_e, k_e) \in \Theta$. We consider a set of $\theta_e$ parameters uniformly sampled from the environment distribution $Q$ on $\Theta$.

**Navier-Stokes (NS).**    We consider the Navier-Stokes PDE for incompressible flows:

$$\partial w/\partial t = -v \cdot \nabla w + \nu \Delta w + \xi \qquad \nabla \cdot v = 0$$

where $v$ is the velocity field, $w = \nabla \times v$ is the vorticity, both $v, w$ lie in a spatial domain $S$ with periodic boundary conditions, $\nu$ is the viscosity and $\xi$ is the constant forcing term in the domain $S$. The discretized state at time $t$ is the vorticity $x_t^e = w_t^e \in \mathbb{R}^{32^2}$. Note that $v$ is already contained in $w$. We fix $\nu = 10^{-3}$ across the environments. We sample the initial conditions $w_0^e \sim P_0$ as in [19]. An environment $e$ is defined by its forcing term $\xi_e \in \Theta_\xi$. We uniformly sampled a set of forcing terms from $Q$ on $\Theta_\xi$.

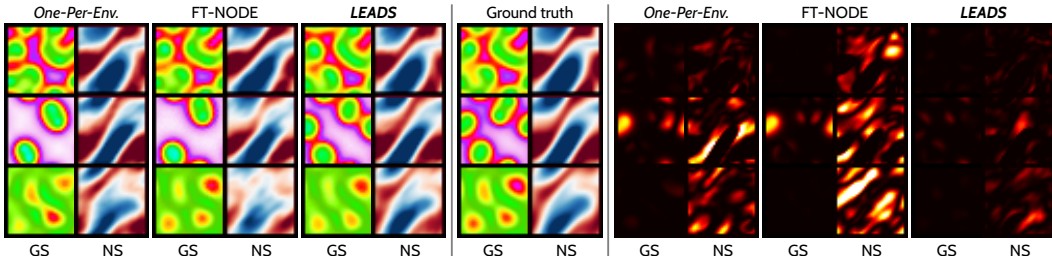

Figure 2: Left: final states for GS and NS predicted by the two best baselines (*One-Per-Env.* and FT-NODE) and *LEADS* compared with ground truth. Different environment are arranged by row (3 in total). Right: the corresponding MAE error maps, the scale of the error map is [0, 0.6] for GS, and [0, 0.2] for NS; darker is smaller. (See Sup. D for full sequences)

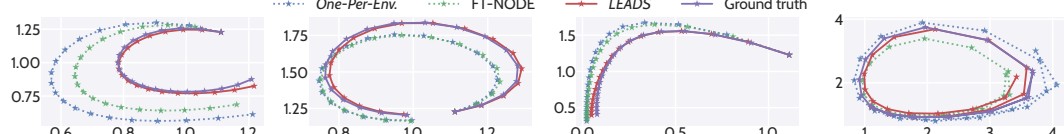

Figure 3: Test predicted trajectories in phase space with two baselines (*One-Per-Env.* and FT-NODE) and *LEADS* compared with ground truth for LV for 4 envs., one per figure from left to right. Quantity of the prey $u$ and the predator $v$ respectively on the horizontal and the vertical axis. Initial state is the rightmost end-point of the figures and it is common to all the trajectories.

**Datasets.** For training, we create two datasets for LV by simulating trajectories of $K = 20$ successive points with temporal resolution $\Delta t = 0.5$. We use the first one as a set of training dynamics to validate the *LEADS* framework. We choose 10 environments and simulate 8 trajectories (thus corresponding to $n = 8 \cdot K$ data points) per environment for training. We can then easily control the number of data points and environments in experiments by taking different subsets. The second one is used to validate the improvement with *LEADS* while training on novel environments. We simulate 1 trajectory ($n = 1 \cdot K$ data points) for training. We create two datasets for further validation of *LEADS* with GS and NS. For GS, we simulate trajectories of $K = 10$ steps with $\Delta t = 40$. We choose 3 parameters and simulate 1 trajectory ($n = 1 \cdot K$ data points) for training. For NS, we simulate trajectories of $K = 10$ steps with $\Delta t = 1$. We choose 4 forcing terms and simulate 8 trajectories ($n = 8 \cdot K$ states) for training. For test-time evaluation, we create for each equation in each environment a test set of 32 trajectories ($32 \cdot K$) data points. Note that every environment dataset has the same number of trajectories and the initial conditions are fixed to equal values across the environments to ensure that the data variations only come from the dynamics themselves, i.e. for the $i$-th trajectory in $\hat{\mathcal{P}}_e$, $\forall e, x_0^{e,i} = x_0^i$. LV and GS data are simulated with the DOPRI5 solver in NumPy [10, 13]. NS data is simulated with the pseudo-spectral method as in [19].

### 4.2 Experimental settings and baselines

We validate *LEADS* in two settings: in the first one all the environments in $E$ are available at once and then $f$ and all the $g_e$s are all trained on $E$. In the second one, training has been performed on $E$ as before, and we consider a novel environment $e' \notin E$: the shared term $f$ being kept fixed, the approximating function $f_{e'} = f + g_{e'}$ is trained on the data from $e'$ (i.e. only $g_{e'}$ is modified).

**All environments available at once.** We introduce five baselines used for comparing with *LEADS*: (a) *One-For-All*: learning on the entire dataset $\hat{\mathcal{P}}$ over all environments with the sum of a pair of NNs $f + g$, with the standard ERM principle, as in [2]. Although this is equivalent to use only one function $f$, we use this formulation to indicate that the number of parameters is the same for this experiment and for the *LEADS* ones. (b) *One-Per-Env.*: learning a specific function for each dataset $\hat{\mathcal{P}}_e$. For the same reason as above, we keep the sum formulation $(f + g)_e$. (c) Factored Tensor RNN or **FT-RNN** [33]: it modifies the recurrent neural network to integrate a one-hot environment code into each linear transformation of the network. Instead of being encoded in a separate function $g_e$ like in *LEADS*, the environment appears here as an extra one-hot input for the RNN linear transformations. This can be implemented for representative SOTA (spatio-)temporal predictors such as GRU [8] or PredRNN [35]. (d) **FT-NODE**: a baseline for which the same environment encoding as FT-RNN is incorporated in a Neural ODE [7]. (e) Gradient-based Meta Learning or **GBML-like** method: we propose a GBML-like baseline which can directly compare to our framework. It follows the

Table 1: Results for LV, GS, and NS datasets, trained on $m$ envs. with $n$ data points per env.

| Method | LV ($m = 10, n = 1 \cdot K$) | | GS ($m = 3, n = 1 \cdot K$) | | NS ($m = 4, n = 8 \cdot K$) | |
|---|---|---|---|---|---|---|
| | MSE train | MSE test | MSE train | MSE test | MSE train | MSE test |
| *One-For-All* | 4.57e-1 | 5.08±0.56 e-1 | 1.55e-2 | 1.43±0.15 e-2 | 5.17e-2 | 7.31±5.29 e-2 |
| *One-Per-Env.* | 2.15e-5 | 7.95±6.96 e-3 | 8.48e-5 | 6.43±3.42 e-3 | 5.60e-6 | 1.10±0.72 e-2 |
| FT-RNN [33] | 5.29e-5 | 6.40±5.69 e-3 | 8.44e-6 | 8.19±3.09 e-3 | 7.40e-4 | 5.92±4.00 e-2 |
| FT-NODE | 7.74e-5 | 3.40±2.64 e-3 | 3.51e-5 | 3.86±3.36 e-3 | 1.80e-4 | 2.96±1.99 e-2 |
| GBML-like | 3.84e-6 | 5.87±5.65 e-3 | 1.07e-4 | 6.01±3.62 e-3 | 1.39e-4 | 7.37±4.80 e-3 |
| *LEADS no min.* | 3.28e-6 | 3.07±2.58 e-3 | 7.65e-5 | 5.53±3.43 e-3 | 3.20e-4 | 7.10±4.24 e-3 |
| **LEADS** (Ours) | 5.74e-6 | **1.16±0.99 e-3** | 5.75e-5 | **2.08±2.88 e-3** | 1.03e-4 | **5.95±3.65 e-3** |

principle of MAML [11], by training *One-For-All* at first which provides an initialization near to the given environments like GBML does, then fitting it individually for each training environment. (f) **LEADS no min.**: ablation baseline, our proposal without the $\Omega(g_e)$ penalization. A comparison with the different baselines is proposed in Table 1 for the three dynamics. For concision, we provide a selection of results corresponding to 1 training trajectory per environment for LV and GS and 8 for NS. This is the minimal training set size for each dataset. Further experimental results when varying the number of environments from 1 to 8 are provided in Fig. 4 and Table S3 for LV.

**Learning on novel environments.** We consider the following training schemes with a pre-trained, fixed $f$: (a) **Pre-trained-$f$-Only**: only the pre-trained $f$ is used for prediction; a sanity check to ensure that $f$ cannot predict in any novel environment without further adaptation. (b) **One-Per-Env.**: training from scratch on $\{\hat{\mathcal{P}}_{e'}\}$ as *One-Per-Env.* in the previous section. (c) **Pre-trained-$f$-Plus-Trained-$g_e$**: we train $g$ on each dataset $\hat{\mathcal{P}}_{e'}$ based on pre-trained $f$, i.e. $f + g_{e'}$, leaving only $g_{e'}$s adjustable. We compare the test error evolution during training for 3 schemes above for a comparison of convergence speed and performance. Results are given in Fig. 5.

### 4.3 Experimental results

**All environments available at once.** We show the results in Table 1. For LV systems, we confirm first that the entire dataset cannot be learned properly with a single model (*One-For-All*) when the number of environments increases. Comparing with other baselines, our method *LEADS* reduces the test MSE over 85% w.r.t. *One-Per-Env.* and over 60% w.r.t. *LEADS no min.*, we also cut 50%-75% of error w.r.t. other baselines. Fig. 3 shows samples of predicted trajectories in test, *LEADS* follows very closely the ground truth trajectory, while *One-Per-Env.* under-performs in most environments. We observe the same tendency for the GS and NS systems. The error is reduced by: around 2/3 (GS) and 45% (NS) w.r.t. *One-Per-Env.*; over 60% (GS) and 15% (NS) w.r.t. *LEADS no min.*; 45-75% (GS) and 15-90% (NS) w.r.t. other baselines. In Fig. 2, the final states obtained with *LEADS* are qualitatively closer to the ground truth. Looking at the error maps on the right, we see that the errors are systematically reduced across all environments compared to the baselines. This shows that *LEADS* accumulates less errors through the integration, which suggests that *LEADS* alleviates overfitting.

We have also conducted a larger scale experiment on LV (Fig. 4) to analyze the behavior of the different training approaches as the number of environments increases. We consider three models *One-For-All*, *One-Per-Env.* and *LEADS*, 1, 2, 4 and 8 environments, and for each such case, we have 4 groups of curves, corresponding to 1, 2, 4 and 8 training trajectories per environment. We summarize the main observations. With *One-For-All* (blue), the error increases as the number of environments increases: the dynamics for each environment being indeed different, this introduces an increasingly large bias, and thus the data cannot be fit with one single model. The performance of *One-Per-Env.* (in red), for which models are trained independently for each environment, is constant as expected when the number of environments changes.

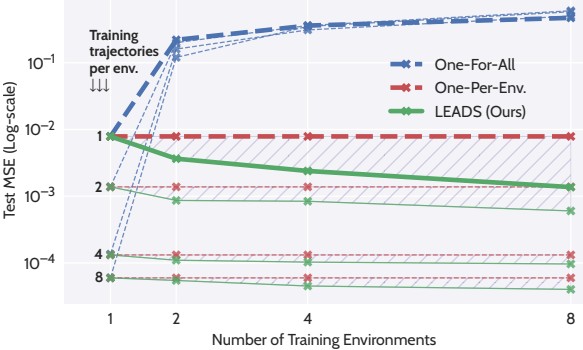

Figure 4: Test error for LV w.r.t. the number of environments. We apply the models in 1 to 8 environments. 4 groups of curves correspond to models trained with 1 to 8 trajectories per env. All groups highlight the same tendencies: increasing *One-For-All*, stable *One-Per-Env.*, and decreasing *LEADS*. More results of baselines methods in Sup. D.

*LEADS* (green) circumvents these issues and shows that the shared characteristics among the environments can be leveraged so as to improve generalization: it is particularly effective when the number of samples per environment is small. (See Sup. D for more details on the experiments and on the results).

**Learning on novel environments.** We demonstrate how the pre-trained dynamics can help to fit a model for novel environments. We took an $f$ pre-trained by *LEADS* on a set of LV environments. Fig. 5 shows the evolution of the test loss during training for three systems: a $f$ function pre-trained by *LEADS* on a set of LV training environments, a $g_e$ function trained from scratch on the new environment and *LEADS* that uses a pre-trained $f$ and learns a $g_e$ residue

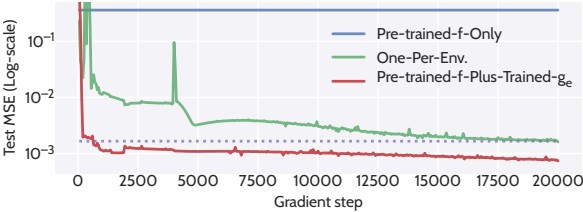

Figure 5: Test error evolution during training on 2 novel environments for LV.

on this new environment. *Pre-trained-f-Only* alone cannot predict in any novel environments. Very fast in the training stages, *Pre-trained-f-Plus-Trained-$g_e$* already surpasses the best error of the model trained from scratch (indicated with dotted line). Similar results are also observed with the GS and NS datasets (cf. Sup. D, Table S5). These empirical results clearly show that the learned shared dynamics accelerates and improves the learning in novel environments.

## 4.4 Training and implementation details

**Discussion on trajectory-based optimization.** Solving the learning problem Eq. 2 in our setting, involves computing a trajectory loss (integral term in Eq. 4). However, in practice, we do not have access to the continuous trajectories at every instant $t$ but only to a finite number of snapshots for the state values $\{x_{k\Delta t}\}_{0 \leq k \leq \frac{T}{\Delta t}}$ at a temporal resolution $\Delta t$. From these discrete observed trajectories, it is still possible to recover an approximate derivative $d_{k\Delta t}^{\Lambda} \simeq \frac{dx_{k\Delta t}}{dt}$ using a numerical scheme $\Lambda$. The integral term for a given sample in the objective Eq. 4 would then be estimated as $\sum_{k=1}^{K} \|d_{k\Delta t}^{\Lambda} - (f + g_e)(x_{\Delta tk})\|^2$. This is not the best solution and we have observed much better prediction performance for all models, including the baselines, when computing the error directly on the states, using an integral formulation $\sum_{k=1}^{K} \|x_{(k+1)\Delta t} - \tilde{x}_{(k+1)\Delta t}\|^2$, where $\tilde{x}_{(k+1)\Delta t}$ is the solution given by a numerical solver approximating the integral $x_{k\Delta t} + \int_{k\Delta t}^{(k+1)\Delta t}(f + g_e)(\tilde{x}_s)ds$ starting from $x_{k\Delta t}$. Comparing directly in the state space yields more accurate results for prediction as the learned network tends to correct the solver's numerical errors, as first highlighted in [37].

**Calculating $\Omega$.** Given finite data and time, the exact infinity norm and Lipschitz norm are both intractable. We opt for more practical forms in the experiments. For the infinity norm, we chose to minimize the empirical norm of the output vectors on known data points, this choice is motivated in Sup. C. In practice, we found out that dividing the output norm by its input norm works better: $\frac{1}{n} \sum_{i,k} \|g_e(x_{k\Delta t}^{e,i})\|^2 / \|x_{k\Delta t}^{e,i}\|^2$, where the $x_{k\Delta t}^{e,i}$ are known states in the training set. For the Lipschitz norm, as suggested in [5], we optimize the sum of the spectral norms of the weight at each layer $\sum_{l=1}^{D} \|W_l^{g_e}\|^2$. We use the power iteration method in [25] for fast spectral norm approximation.

**Implementation.** We used 4-layer MLPs for LV, 4-layer ConvNets for GS and Fourier Neural Operator (FNO) [19] for NS. For FT-RNN baseline, we adapted GRU [8] for LV and PredRNN [35] for GS and NS. We apply the Swish function [31] as the default activation function. Networks are integrated in time with RK4 (LV, GS) or Euler (NS), using the basic back-propagation through the internals of the solver. We apply an exponential Scheduled Sampling [17] with exponent of 0.99 to stabilize the training. We use the Adam optimizer [15] with the same learning rate $10^{-3}$ and $(\beta_1, \beta_2) = (0.9, 0.999)$ across the experiments. For the hyperparamters in Eq. 8, we chose respectively $\lambda = 5 \times 10^3, 10^2, 10^5$ and $\alpha = 10^{-3}, 10^{-2}, 10^{-5}$ for LV, GS and NS. All experiments are performed with a single NVIDIA Titan Xp GPU.

## 5 Related work

Recent approaches linking invariances to Out-of-Distribution (OoD) Generalization, such as [1, 16, 34], aim at finding a single classifier that predicts well invariantly across environments with the power of extrapolating outside the known distributions. However, in our dynamical systems context, the optimal regression function should be different in each environment, and modeling environment

bias is as important as modeling the invariant information, as both are indispensable for prediction. Thus such invariant learners are incompatible with our setting. Meta-learning methods have recently been considered for dynamical systems as in [11, 18]. Their objective is to train a single model that can be quickly adapted to a novel environment with a few data-points in limited training steps. However, in general these methods do not focus on leveraging the commonalities and discrepencies in data and may suffer from overfitting at test time [24]. Multi-task learning [38] seeks for learning shared representations of inputs that exploit the domain information. Up to our knowledge current multi-task methods have not been considered for dynamical systems. [33] apply multi-task learning for interactive physical environments but do not consider the case of dynamical systems. Other approaches like [39, 28] integrate probabilistic methods into a Neural ODE, to learn a distribution of the underlying physical processes. Their focus is on the uncertainty of a single system. [37] consider an additive decomposition but focus on the combination of physical and statistical components for a single process and not on learning from different environments.

## 6   Discussions

**Limitations**   Our framework is generic and could be used in many different contexts. On the theoretical side, the existence and uniqueness properties (Prop. 1) rely on relatively mild conditions covering a large number of situations. The complexity analysis, on the other side, is only practically relevant for simple hypothesis spaces (here linear), and then serves for developing the intuition on more complex spaces (NNs here) where bounds are too loose to be informative. Another limitation is that the theory and experiments consider deterministic systems only: the experimental validation is performed on simulated deterministic data. Note however that this is the case in the vast majority of the ML literature on ODE/PDE spatio-temporal modeling [30, 20, 19, 37]. In addition, modeling complex dynamics from real world data is a problem by itself.

**Conclusion**   We introduce *LEADS*, a data-driven framework to learn dynamics from data collected from a set of distinct dynamical systems with commonalities. Experimentally validated with three families of equations, our framework can significantly improve the test performance in every environment w.r.t. classical training, especially when the number of available trajectories is limited. We further show that the dynamics extracted by *LEADS* can boost the learning in similar new environments, which gives us a flexible framework for generalization in novel environments. More generally, we believe that this method is a promising step towards addressing the generalization problem for learning dynamical systems and has the potential to be applied to a large variety of problems.

## Acknowledgements

We acknowledge financial support from the ANR AI Chairs program DL4CLIM ANR-19-CHIA-0018-01.

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
