# OpenReview forum: "LEADS: Learning Dynamical Systems that Generalize Across Environments"
_NeurIPS.cc/2021/Conference — NeurIPS 2021 Poster_

### Official Review · Reviewer_35iL · 2021-07-15

**Rating:** 7
**Confidence:** 3

**Summary:**

This paper presents an approach called LEADS to learn *decomposed* dynamics models of system instances ("environments") from a family of systems, e.g., Lotka-Volterra systems with randomly sampled parameters. The main idea is to additively decompose the dynamics into a shared component $f$ which does not depend on the specific environment and an environment-specific component $g_e$. It is shown both theoretically and experimentally that it is crucial to limit the complexity of $g_e$. This allows to adapt to new, unseen environments by leveraging "knowledge" absorbed in $f$ and additionaly using only a few interactions with the new environment to learn a new $g_{e'}$. The performed experiments qualitatively and quantitatively confirm the hypothesis that the proposed algorithm LEADS is more accurate than training a single model for all environments and even a single model for each environment.



**Limitations And Societal Impact:**

* The authors clearly mention the limitations of their work.

* Societal impacts are not explicitly addressed. In my opinion, this work does not exhibit extraordinary societal impacts in addition to those induced by basic reasearch on dynamics modelling methods in general.

**Main Review:**

**Originality**

The paper suggests an additive decomposition of environment dynamics into a *shared* and *environment-specific* part. To my understanding, the main methodological novelty is that *both* parts are learned while a minimization objective on the environment-specific part is imposed. With regard to this, some related work should be discussed more precisely in my opinion:

  * Another instance of dynamics models which are decomposed into a *shared* part and an *environment-specific* part are "context-aware dynamics models" or "context-conditional dynamics models", e.g. by (Lee et al., 2020). I think the paper would benefit from mentioning these approaches.

  * An additive decomposition of a gradient field of an ODE has been described in (Yin et al. 2021), who also highlight the need for a minimization objective on one of the components (Yin et al., 2021, eq. 2) and show existence and uniqueness properties (Yin et al., 2021, Prop. 1+2). I think the paper of Yin et al. (2021) should be, due to this overlap, discussed in more detail in the paper under review.

  * I think the statement in l. 75/76 "Up to our knowledge, this is the first time it is addressed from an ML viewpoint" needs clarification. To me it is not immediately clear what is meant by "this". Has the problem/question already been addressed from a different, non-ML viewpoint?


**Quality**

* The chosen approach is well-motivated by mathematical derivations, e.g. regarding limiting the complexity of the environment-specific dynamics components.

* All experiments are performed on a set of ODE/PDE dynamics (Lotka-Volterra, Gray-Scott and Navier-Stokes), showing that the reported effects are valid for multiple systems.

* Reasonable ablation and comparison experiments are performed highlighting the importance of the minimization objective.


**Clarity**

* The paper is well-structured and clearly written. The relevant research questions are clearly outlined.

* The figures nicely illustrate the results obtained using the method and the main statements of the paper. Some minor comments:
  * Fig. 2: I would appreciate a colorbar for the MSE error maps to at least estimate the quantitative difference in the predictions between LEADS and its ablations / baseline methods.
  * Fig. 4: Coordinate axes ('lines') at the bottom and on the right (including 'ticks') would help to immediately see that the y coordinate of the curves is not determined by "Training trajectories per env.", which I first falsely assumed to be a second "axis" on the left.
  * For consistency, I suggest to introduce coordinate axes also in Fig. 1, 3, 5.

* The derivations and experiments for the 'simplified' linear case help in understanding the mechanics of the method before considering the more-complicated NN models.


**Significance and conclusion**:
In my opinion the proposed idea of decomposing dynamics into a "general" component and an "environment-specific" component is very interesting and could, for example, help the community to build more sample-efficient model-based RL algorithms which are able to quickly adapt to changing environments. Although the idea of additive decomposition seems simple, it requires significant theoretical effort to concretize this idea, especially regarding the required complexity regularization. This paper provides a strong theoretical formulation and experimental evaluation of the proposed method. Noting that I haven't checked the mathematical derivations in detail, I vote for acceptance.


**References:**

 (Lee et al., 2020):
 Kimin Lee, Younggyo Seo, Seunghyun Lee, Honglak Lee, Jinwoo Shin
 "Context-aware Dynamics Model for Generalization in Model-Based Reinforcement Learning"
 ICML2020, https://arxiv.org/abs/2005.06800

 (Yin et al., 2021):
 Yuan Yin, Vincent Le Guen, Jérémie Donà, Emmanuel de Bézenac, Ibrahim Ayed, Nicolas Thome, Patrick Gallinari
 "Augmenting Physical Models with Deep Networks for Complex Dynamics Forecasting"
 ICLR2021, https://arxiv.org/abs/2010.04456

**Time Spent Reviewing:**

5

---

> ### Author Response · Authors · 2021-08-10
> **Response to Reviewer 35iL**
>
> We thank the reviewer for the positive comments and constructive remarks and  questions. We respond to these points in detail below.
>
> - **Contextual dynamical models**.
>
>   We do agree, these models share a similar objective with LEADS for generalization in new environments. We will add a more detailed discussion w.r.t. meta learning and physically meaningful approaches and quote corresponding papers. With a similar adaptation to different environments objective, recent works such as (Zintgraf et al., 2019, Sitzmann et al., 2020), propose to use Hyper-Networks (Ha et al., 2017) or similar ideas to extract informative and distinctive representations in different environments. This too will also be added to the discussion.
>
> - **Discussion on (Yin et al., 2021)**.
>
>   Thank you for bringing up this issue; a clear discussion on this paper is lacking and will be updated in the manuscript.
>
>   Although indeed sharing some similar technical ideas with (Yin et al., 2021), including the additive decomposition along with a penalty term, our framework is clearly distinct in many aspects:
>   - (a) The objectives and tasks are different: LEADS explores the generalization in multiple dynamical systems, while (Yin et al., 2021) focuses on the introduction of physical background in statistical models and proposes a framework to combine physical modeling with neural networks for a single dynamical system.
>   - (b) The assumptions are different: LEADS is a pure ML framework, while (Yin et al., 2021) suppose that a part of the model is defined by some prior knowledge of the underlying physics, their goal being to use this prior knowledge in order to improve generalization while no such assumption is made in our work.
>   - (c) The theories and the corresponding technical aspects are different: The existence and unicity in LEADS mainly depend on the choice of the capacity control \Omega, while (Yin et al., 2021)’s theoretical results depend on the geometry of the physical prior.
>
> - **Has the problem/question already been addressed from a different, non-ML viewpoint?**
>
>   After re-reading, we agree that our sentence is far too vague and it will be stated more precisely.
>
>   We only wanted to state that learning to generalize from several dynamical systems as described in our context has not yet been considered before in the ML community. However, since the idea of improving a general system (in our case the shared component) with a complementary model that takes into consideration some additional factor is quite general in engineering, we did not want to claim that the novelty extended beyond the ML community. Besides we are not aware of similar attempts in other communities, but our expertise is of course limited.
>
> - We will add the scales of error maps in Figure 2: the error map scale for GS is [0, 0.6] (dark->bright), and the scale for NS is [0, 0.2]. We will also refine the figures according to the remarks when possible.
>
> References:
>
> - (Ha et al., 2017) David Ha, Andrew M. Dai, Quoc V. Le: HyperNetworks. ICLR 2017
> - (Zintgraf et al., 2019) Luisa M Zintgraf, Kyriacos Shiarlis, Vitaly Kurin, Katja Hofmann, Shimon Whiteson: Fast Context Adaptation via Meta-Learning, ICML 2019
> - (Sitzmann et al, 2020) Vincent Sitzmann, Julien N. P. Martel, Alexander W. Bergman, David B. Lindell, Gordon Wetzstein: Implicit Neural Representations with Periodic Activation Functions, NeurIPS 2020

---

### Official Review · Reviewer_hYVh · 2021-07-16

**Rating:** 6
**Confidence:** 3

**Summary:**

This proposes a simple but effective method to learn a dynamics prediction model that can generalize different environments. The authors also provide theoretical and experimental results to demonstrate the superiority of the proposed method over baselines.

**Limitations And Societal Impact:**

See the section "Main Review"

**Main Review:**

Advantages:
   1. Learning a dynamics prediction model that can generalize across different environments is important and practical for many real-world applications, and the authors provide a simple and intuitive method to solve it with theoretical guarantees.
   2. The experimental results shown in the main paper and Appendix are satisfactory.

Disadvantage:
  1. I notice that the proposed method can only be applied to deterministic dynamics prediction, which limits its potential application scenarios.
  2. The hyperparameters \lambda and \alpha varies significantly across different environments. Can the authors provide the results about the sensitivity of hyperparameters?

**Time Spent Reviewing:**

3

---

> ### Author Response · Authors · 2021-08-10
> **Response to Reviewer hYVh**
>
> We thank the reviewer for the constructive remarks and questions.
>
> - **Deterministic dynamics limitation**.
>
>   In the real-world, a great number of dynamical phenomena can be described in a deterministic way, e.g. liquid motions in physics, chemical reactions in chemistry, species interaction in ecosystems, epidemy dynamics, cardiac conduction in medicine. This is the case e.g. for all the situations evaluated in our experiments. We focus in this work on learning in different environments and although we agree that considering only deterministic dynamics is somewhat of a limitation, this already allows us to show the effectiveness of our framework for a large range of phenomena. We believe non-deterministic dynamics or any additional stochastic component, other than the one induced by the different environments, brings an additional level of complexity but could nonetheless be considered using an adaptation of our framework, using stochastic differential equations. However, although very interesting, we believe the treatment of this problem is out of the scope of the current work.
>
> - **Sensitivity of the hyperparameters**.
>
>   As usual, the hyperparameters need to be tuned for each considered set of systems. We therefore chose the hyperparameters using standard cross-validation techniques. We did not conduct a systematic sensitivity analysis. In practice, we found that: (a) if the regularization term is too large w.r.t. the trajectory loss, the model cannot fit the trajectories, and (b) if the regularization term is too small, the performance is similar to LEADS no min. The candidate hyperparameters are defined on a very sparse grid, for example (1e3, 1e4, 1e5, 1e6) for lambda and (1e-2, 1e-3, 1e-4, 1e-5) for alpha. We will update these details in the manuscript when possible.

---

### Official Review · Reviewer_27Ns · 2021-07-16

**Rating:** 6
**Confidence:** 2

**Summary:**

The paper proposes an algorithm which learns to model dynamical systems
consisting of multiple environments that exhibit shared and
environment-specific dynamics. The key idea is to learn separate approximation
functions for shared and environment-specific components. To prevent a
trivial solution that circumvents the approximation of shared dynamics, the
proposed objective regularizes environment-specific terms.

On the theoretical side, the paper quantifies the improvement in sample
complexity for novel environments in a simplified (linear) setting for which a
suitable regularization term is derived. Empirically, the method is evaluated
on three datasets with nonlinear dynamics (all datasets have shared and
environment-specific components) and the results support the claims about
improved generalization on known and novel environments, especially when data
is scarce (i.e., limited number of trajectories).


**Limitations And Societal Impact:**



**Main Review:**

Even though the paper is outside my domain of expertise, it was mostly easy
to follow, due to the clear and unambiguous writing style and notation, as well
as the excellent explanations throughout the paper. The authors are transparent
about the limitations of the proposed approach. The distinction from previous
work in related fields appears somewhat superficial (see Comments and Questions).

The presented empirical results show consistent and significant improvements
compared to all baselines, supporting the claims about improved generalization
on known and novel environments. However, I think that certain ablations (see
Comments and Questions) could provide a more complete picture of the results
and the suitability of the employed heuristics (e.g., the regularization term
for neural networks). Due to my lack of domain knowledge, I cannot evaluate the
originality and significance of this work in the context of dynamical systems.


**Comments and Questions:**

- Could you give an example for a real-world dynamical system that exhibits the
  properties assumed in this work (i.e., shared and
  environment-specific mechanisms across a bunch of environments)? Is there a
  particular system that you imagine in practice or are the three presented
  datasets the closest examples that you have for such a dynamical system?

- In the related work section, the distinction from work in related fields
  seems somewhat hand-wavy. With respect to meta-learning approaches, even if
  they do not focus on generalization on training environments, that does not
  imply that they do not generalize well on training environments. Even if
  meta-learning approaches or ODE2VAE (Yildiz et al., 2019) focus on a single
  model or a single dynamical system, it could still be interesting to include
  the methods as a benchmark.  For example, does their test error also increase
  with the number of training environments (similar to *One-For-All* in Figure
  4)?

- Did you consider running an ablation for the hyperparameter lambda? It could
  be instructive to see the generalization performance for different
  values of lambda.

- From Appendix B.4 and Appendix C, I understand that the regularizer from
  Equation (7) is approximated via heuristics that introduce additional
  hyperparameters. For example, the choice of $p$-norm (Table S4) seems to have
  a non-negiglible effect on the results of the LV experiment. In light of this
  variablility, did you consider to include the results for different $p$-norms
  in Table 1 directly?

- For Figure 4, do you also have results for FT-RNN and FT-ODE? It would be
  interesting to see how robust the methods perform with respect to the number
  of training environments.

- Lines 103f: could you point out the source or theorem for why the additive
  hypothesis is not restrictive given functional vector spaces?

**Time Spent Reviewing:**

6

---

> ### Author Response · Authors · 2021-08-10
> **Response to Reviewer 27Ns**
>
> We thank the review for the positive comments and constructive questions and remarks. We provide some additional experimental results to answer some of your comments. Detailed below are our responses.
>
> - **Example for real-world dynamical systems that correspond to our setting.**
>
>   Let us consider two examples briefly introduced in the paper. Note that the problem  of generalization to different environments raised in this work stems from attempting to apply ML to these precise domains.
>
>   - (a) Geophysics: for example, the underlying ocean dynamics follow NS equations, but local forces applied to these systems are different in different spatial regions (e.g. Coriolis), and thus the behavior of the system is different. Each location or broad spatial area could then correspond to an environment.
>
>     However, probably more important is the difference between physical and statistical models.  Physical models postulate general laws and in geophysics, their parameters are usually estimated and used at a global level (e.g. at ocean scale). They are often further calibrated via data assimilation (Abarbanel et al., 2018, Brajard et al., 2020) by combining the physical model with observed information so as to regularly update the parameters of the model when observations become available (this is often performed via some form of Kalman filtering).
>
>     In ML we face a different difficulty. We adopt an agnostic viewpoint (at least this is what we do in this paper). We then learn from samples corresponding to observations (simulations in our paper). Whatever the number of physical observations is, we will not be able to cover the range of situations that occur in nature. In geophysic, most often, the complexity of the underlying system is such that even with large amounts of observations, e.g. via satellite measures, we face a scarcity problem and we are very far from covering the range of possible situations. In an ML setting, typically, one will learn from limited spatial or temporal situations, e.g. by selecting some spatial regions of interest, and there is no guarantee that the trained model will generalize to other spatial/ time contexts: we learn correlations and not the laws of nature - and most often the trained model does not generalize. Clearly, physicists will not buy models without generalization guarantees. We will then have to find an approach in order to help the model generalize to new environments. We propose such a mechanism in the paper. For geophysics, each spatial/ temporal context could then be considered as an environment.
>
>   - (b) Medicine: There are several cases in the medical domain, such as for personalized medicine that motivate the framework. For example in cardiac electrophysiology (Fresca et al., 2020, Giffard-Roisin et al., 2017), one often postulates a simplified electrical activity propagation law. This law is inaccurate and should be complemented for more precise modeling, and more important for our objective, each patient will have its specificities and require a specific adaptation. This means that if one learns from a subset of patients, there is no guarantee that this will generalize to others. In this case each patient will represent an environment.
>
> - **Distinction with meta-learning**.
>
>   We agree with your remark. Meta-learning covers a broad range of ideas and methods and indeed our setting and meta-learning have a similar general objective concerning generalization for different contexts. More precisely, LEADS shares a similar objective for generalization with gradient-based meta learning (GBML) methods. The difference is that, to fill the gap between the function shared across environments and each target environment, LEADS uses a regularized environment-specific term, while GBML methods use gradient descent within a single global model. Also, current GBML methods require a query/support set separation in the training data to perform meta training, which makes a proper comparison with our framework not so direct. Despite these differences, we will try to propose a GBML-like method that can be directly compared to our framework, by training One-for-All at first which provides an initialization near to the given environments like GBML does then fitting it individually for each training environment. Time was too short during the rebuttal, but we will try to address this issue during the discussion period.
>
> - **Hyperparameter sensitivity**.
>
>   The hyperparameters need to be tuned for each considered set of systems. We chose the hyperparameters using the standard cross-validation techniques, but we did not perform a systematic sensitivity analysis. In practice, we found that: (a) if the regularization term is too large in the face of trajectory loss, the model cannot fit the trajectories, and (b) if the regularization term is too small, the performance is similar to LEADS no min. The candidate hyperparameters are defined on a very sparse grid, for example (1e3, 1e4, 1e5, 1e6) for lambda and (1e-2, 1e-3, 1e-4, 1e-5) for alpha. We will update these details in the manuscript when possible.
>
> - **Choice of $p$-norm**.
>
>   All proposed $p$-norms are in theory equivalent to achieve the same infinity function norm objective in (7) on the space of training data. However, they perform very differently in practice, because the gradient of each choice is different. When p is too large, only few data points are used to optimize (7), which is inefficient and may disturb the trajectory fitting. We therefore treat the p-norm as a hyperparameter, and we choose it empirically for all experiments.
>
> - **We provide below new results corresponding to Fig 4 for FT-RNN and FT-ODE**.
>
>   We conducted some extra experiments for FT-RNN, and FT-NODE. We observe that LEADS always outperforms two baselines and LEADS generalizes better when more environments are used for training, which is not always valid for FT-RNN and FT-NODE.
>
> |                           |                    |  1 env*            |  2 env               | 4 env                | 8 env              |
> |----------------------| ----------------|---------------------|----------------------|----------------------|---------------------|
> |  1 traj per env    |    FT-RNN   | 4.02+-3.17 e-2 | 1.62+-1.14 e-2 | 1.62+-1.40 e-2 | 1.08+-1.03 e-2|
> |                           |    FT-NODE | 7.87+-7.54 e-3 | 7.63+-5.84 e-3 | 4.18+-3.77 e-3 | 4.92+-4.19 e-3|
> |                           |    LEADS     | 7.87+-7.54 e-3 | **3.65+-2.99 e-3** | **2.39+-1.83 e-3** | **1.37+-1.14 e-3**|
> |  2 traj per env    |    FT-RNN   | 7.20+-7.12 e-2 | 2.72+-4.00 e-2 | 1.69+-1.57 e-3 | 1.38+-1.25 e-3|
> |                           |    FT-NODE | 1.38+-1.61 e-3 | 9.02+-8.81 e-3 | 1.11+-1.05 e-3 | 1.00+-0.95 e-3|
> |                           |    LEADS     | 1.38+-1.61 e-3 | **8.65+-9.61 e-4** | **8.40+-9.76 e-4** | **6.02+-6.12 e-4**|
> |  4 traj per env    |    FT-RNN    | 8.69+-8.36 e-4 | 3.39+-3.38 e-4 | 3.02+-1.50 e-4 | 2.26+-1.45 e-4|
> |                           |    FT-NODE  | 1.36+-1.25 e-4 | 1.74+-1.65 e-4 | 1.78+-1.71 e-4 | 1.39+-1.20 e-4|
> |                           |    LEADS      | 1.36+-1.25 e-4 | **1.10+-0.92 e-4** | **1.03+-0.98 e-4** | **9.66+-9.79 e-5**|
> |  8 traj per env    |    FT-RNN    | 2.09+-1.73 e-4 | 1.18+-1.16 e-4 | 1.13+-1.13 e-4 | 9.13+-8.31 e-5|
> |                           |    FT-NODE  | 5.98+-5.13 e-5 | 6.91+-4.46 e-5 | 7.82+-6.95 e-5 | 6.88+-6.39 e-5|
> |                           |    LEADS      | 5.98+-5.13 e-5 | **5.47+-4.63 e-5** | **4.52+-3.98 e-5** | **3.94+-3.49 e-5**|
>
> \* For results with 1 environment, FT-NODE is reduced to One-Per-Env as LEADS does, we therefore report the same value for both.
>
> - **Could you point out the source or theorem for why the additive hypothesis is not restrictive given functional vector spaces?**
>
>   By assumption, we formulate the task we solve in a functional vector space which is closed under addition by definition. In practice, the functions of this space are approximated by the chosen parametrization and, assuming that the space of parametrized functions is dense in the vector space, there always exists an additive decomposition as well as a set of parameters which approximate its terms.
>
>
> References:
>
> - (Abarbanel et al., 2018) Henry D. I. Abarbanel, Paul J. Rozdeba, Sasha Shirman; Machine Learning: Deepest Learning as Statistical Data Assimilation Problems. Neural Comput 2018; 30 (8): 2025–2055.
> - (Brajard et al., 2020) Julien Brajard, Alberto Carrassi, Marc Bocquet, Laurent Bertino:
> Combining data assimilation and machine learning to emulate a dynamical model from sparse and noisy observations: A case study with the Lorenz 96 model. J. Comput. Sci. 44: 101171 (2020)
> - (Fresca et al., 2020) Fresca, S., Manzoni, A., Dedè, L. and Quarteroni, A. 2020. Deep learning-based reduced order models in cardiac electrophysiology. PLoS ONE, October 2020.
> - (Giffard-Roisin et al., 2017) Sophie Giffard-Roisin, Thomas Jackson, Lauren Fovargue, Jack Lee, Herve Delingette, Reza Razavi, Nicholas Ayache, Maxime Sermesant:
> Noninvasive Personalization of a Cardiac Electrophysiology Model From Body Surface Potential Mapping. IEEE Trans. Biomed. Eng. 64(9): 2206-2218 (2017)

---

> > ### Author Response · Authors · 2021-08-22
> > **New results of the GBML-like method**
> >
> > We provide the new results of the GBML-like method. This method trains at first One-for-All to obtain a network initialization near the given training data as GBML does, then it is fitted individually on the same training data for each training environment.
> >
> > The following table compares the GBML-like method with LEADS in training environments as in Table 1. The test losses of this method are much larger than LEADS while the training loss remains similar.
> >
> > | | LV                 || GS                 || NS      ||
> > |----------------|------------|----------------------|------------|---------------------|-----------|--------------------------|
> > |  | Train MSE  | Test MSE        | Train MSE  | Test MSE        | Train MSE  | Test MSE         |
> > |GBML-like | 3.84e-6 | 5.87+-5.65 e-3 | 1.07e-4 | 6.01+-3.62 e-3 | 1.39e-4 | 7.37+-4.80 e-3|
> > |LEADS    | 5.76e-6 | **1.16+-0.99 e-3** | 5.75e-5 | **2.08+-2.88 e-3** | 1.03e-4 | **5.95+-3.65 e-3**|
> >
> > We also show below the corresponding results for Figure 4. We see that LEADS still always outperforms the GBML-like method and more environments do not always help to improve the generalization of the GBML-like method in all known environments.
> >
> > |       | Test MSE | 1 env\*      | 2 env       | 4 env        | 8 env        |
> > |------------|--------------------|---------------------|----------------------|---------------------|------------------------|
> > | 1 traj per inv  |  GBML-like | 7.87+-7.54 e-3 | 6.32+-5.72 e-2 | 1.44+-0.66 e-1 | 9.85+-8.84 e-3|
> > | |  LEADS   | 7.87+-7.54 e-3 | **3.65+-2.99 e-3** | **2.39+-1.83 e-3** | **1.37+-1.14 e-3** |
> > | 2 traj per inv  |  GBML-like | 1.38+-1.61 e-3 | 9.26+-8.27 e-3 | 1.17+-1.09 e-2 | 1.96+-1.95 e-2|
> > | |  LEADS   | 1.38+-1.61 e-3 | **8.65+-9.61 e-4** | **8.40+-9.76 e-4** | **6.02+-6.12 e-4** |
> > | 4 traj per inv  |  GBML-like | 1.36+-1.25 e-4 | 2.57+-7.18 e-3 | 2.65+-3.26 e-3 | 2.36+-3.58 e-3|
> > | |  LEADS   | 1.36+-1.25 e-4 | **1.10+-0.92 e-4** | **1.03+-0.98 e-4** | **9.66+-9.79 e-5** |
> > | 8 traj per inv  |  GBML-like | 5.98+-5.13 e-5 | 1.02+-1.68 e-4 | 1.41+-2.68 e-4 | 0.99+-1.53 e-4|
> > | |  LEADS   | 5.98+-5.13 e-5 | **5.47+-4.63 e-5** | **4.52+-3.98 e-5** | **3.94+-3.49 e-5** |
> >
> > \* For results with 1 environment, the GBML-like method is reduced to One-Per-Env. as LEADS does; we, therefore, report the same value for both.

---

### Official Review · Reviewer_juRp · 2021-07-19

**Rating:** 7
**Confidence:** 3

**Summary:**

This paper proposes a new framework (LEADS) for learning dynamical systems (ODEs, PDEs) with the aim of generalising across related, but slightly different environments. It uses an additive two-component model, comprising a shared model and an environment-specific model. The paper presents some generic sample complexity results, then looks at the special case of a linear ODE. The paper claims this motivated a choice for an objective in the nonlinear NN case. This is then experimentally investigated in the small data regime, with just a few example trajectories, and just a few different environments. The paper claims the proposed framework significantly outperforms other baselines and ablations.

**Limitations And Societal Impact:**

Some technical limitations were discussed in the work. No societal impact was discussed.

**Main Review:**

**Originality**

I’m not an expert in this area, but the theoretical propositions seem to be generic, from their reference [4], with some mild adaptation to their setting - this doesn’t seem that original in my opinion. Further, they don’t properly reference this with respect to [4], stating only that the theory was inspired by [4]. More detail is needed to properly assess the originality of this section. I would have also liked to see more detail in the related work section. To the best of my knowledge, the proposed method seems to be original.

**Quality**

I couldn’t follow the proof of Prop 2 in supplemental, I don’t understand why the factor of 4/delta appears in c_g term, but not in c_f term. I haven’t tried to verify the proof of other statements. I didn’t understand why the linear case motivated taking equation (7) for the NN case. I would like a scale for the MSE error maps in fig 2 to assess quality. Otherwise, qualitative results in figs2,3 seem good quality. Table 1: Needs more detail on how significance window was calculated. I felt for LV and NS that LEADS wasn’t significantly better than the ablation LEADS no min. For GS, I felt LEADS wasn’t significantly better than FT-NODE (note typo in table 1, FT-ODE -> FT-NODE). To complement fig 5, I would like to see results on GS and NS. I would have also liked to see performance differences in the regime of larger number of environments and trajectories.

**Clarity**

The prose in most of the paper was acceptable, but I found the math difficult to read, especially prop 2-4. Are these propositions really necessary for the main paper? They didn’t help me as a non-expert reader to understand what was going on, or why this was important. All figures would benefit from being larger.

**Significance**

As noted in the limitations section, this only investigated simulated, deterministic data. I found this surprising because when considering generalization, this is typically done in the setting where the data contains some noise, and one tries not to overfit to that noise.
Moreover, considering this paper is using neural networks, which are typically better in the larger data regime, this makes me question the significance of the results, compared to the baselines (especially those that use neural networks).
As also noted in limitations section, the complexity analysis is only relevant in simple hypothesis spaces, such as the linear case. But already this wasn’t enough for the simple NN experiments in this paper.

----

Edit:

Following discussion period, I have now increased my score from 4 to 7, based on author's better explanation of equation (7), and their other proposed changes.

**Time Spent Reviewing:**

5

---

> ### Author Response · Authors · 2021-08-10
> **Response to Reviewer juRp**
>
> We thank the reviewer for the constructive remarks and comments, which we address and detail below. Note that we have also added some complementary experimental results as suggested by your comments.
>
> 1. Theory:
> - **On the originality of our results w.r.t. ref [4].**
>
>   While the general complexity framework is indeed, and as indicated in the paper, inspired from [4], the results cannot be applied directly nor can they be easily adapted as they are developed in [4] for a setting where the models are constructed as compositions, i.e. a common feature extractor mapping inputs into feature space and a classifier for each task mapping features to predictions. We instead consider an additive composition framework and more importantly models of temporal (ODE) or spatio-temporal (PDE) dynamic processes, which require a completely different treatment than classifiers. We thus had to redefine all relevant mathematical objects and rewrite all derivations and proofs, which are substantially different, in order to prove similar complexity bounds for our additive decomposition. This part of the work represented an important effort and was fraught with quite a few difficulties.
>
>   Regarding their significance, those results allow us to obtain bounds which are relatively sharp for simple hypothesis spaces (such as in the linear case). Overall, they serve to guide our intuition in building our framework and provide trends which are coherent with the experimentations. Finally, let us note that the main obstacle to applying those results is the current lack of accurate bounds for the effective capacity of NNs as they are optimized and used in practice. If improved estimates were available then our results would apply to NNs as well as in the linear case.
>
> - **Proof of proposition 2.**
>
>   Factor $4/\delta$ in the $C_G$ term but not in $C_F$: this is simply because the bounds make use of a log-scale. Before applying the log function, $4/\delta$ is indeed a factor for both $C_G$ and $C_F$, after the log it simply appears as a constant additional term. We will update the writing in order to avoid such misunderstanding.
>
> - **Why the linear case motivates the use of eq. (7) for NNs.**
>
>   Our objective through the theoretical analysis is to motivate our strategy for the effective control and adaptation of the model components via relevant generalization bounds and complexity measures. For large overparameterized models like NNs, all the bounds coming from any sample complexity framework are loose. This was a weakness of the first version of this paper. As suggested by reviewers from ICML (see our submission history), we instantiated our theory for a simpler, tractable parametrization, the linear case, where the link between theory (generalization bounds) and our practical framework for controlling the element capacity is clear.  Our strategy being theoretically motivated in this simplified setting, it is then possible to evaluate it empirically for the more complex case of NNs for which the theoretical bounds are not sharp enough. Again, this is due to the lack of precise statistical estimates for the effective sample complexity of NNs, for which the generalization bounds are known to be unrealistic. Finding tight bounds which reflect modern practices is still an open problem. Indeed, our experiments clearly demonstrate that our intuition is correct and allows us to improve the generalization performance in different settings and use cases.
>
> - **Usefulness of theorems 2-4.**
>
>   Theorems 2 and 3 are key pieces in our demonstration. They establish the link between the generalization objective and the optimization problem (3) which implements the intuition of environment-specific minimal correction. Theorem 4 shows that we can actually optimize some values in the theoretical bounds to control the generalization error in practice. Altogether, these results motivate our method of crafting optimizable controls \Omega from theoretical bounds to improve the generalization.
>
> 2. Experiments
>
> - **Significance of the results.**
>
>   The differences between the improvements obtained for the different use cases simply comes from their specific nature, their complexity and the dynamics involved in the use case. If one wants to summarize, throughout the experiments, the test error of LEADS no min is x1.2-x2.6 larger than LEADS, the test error of FT-NODE is x1.8-x4.9 larger than LEADS. Since the use cases have been specifically selected so as to cover a range of situations, we believe that this clearly provides experimental evidence on the ability of LEADS to reduce generalization error in practice.
>
> - **How are the significance windows computed?**
>
>   We report the mean and standard deviation of the trajectory errors.
>
> - **Significance of the results compared to NN baselines.**
>
>   The proposed model and the NN baselines share roughly the same order of complexity in each environment. This makes the comparison fair. Besides, note that for many physical problems, even when the data are plentiful (such as those coming from satellites for earth observation), they only represent a small proportion of the possible situations to be modeled. This is due to the complexity of the underlying system and this effect is often orders of magnitude larger than for current successful applications of NNs. In our case, this means that it is relevant that the data per environment can still not be sufficient to optimally learn a NN for each environment. We proposed the One-Per-Env. baseline to show the suboptimality of such individually trained NNs.
>
> - **Performance differences in different regimes w.r.t the number of environments and trajectories.**
>
>   We performed large scale experiments on the LV dataset to qualify (Figure 4) the trends of the LEADS framework, when trained with different numbers of environments and trajectoires per environment. We observe that the performance gain is more significant when data is scarce per environment. Also, the more environments considered, the larger the test loss reduction is. This is in accordance with the above remark concerning data scarcity for physical systems. We did not perform such systematic tests with the other use cases, because we felt it was enough to demonstrate this tendency on one use case.
>
> - **Results on GS and NS for Fig 5.**
>
>   We show below new results complementing Fig. 5 for the GS and NS datasets, trained on two new environments. We compare here the test MSE at different training iterations. The conclusion remains the same: LEADS improves generalization and accelerates training in novel environments.
>
> **NEW RESULTS for complementing Fig. 5**
>
> |    |Test MSE at gradient step             |    500    |   2500   |  10000 |
> |--------------|---------------------------------------------|------------|-----------|-----------|
> |GS              | Pretrained-$f$-only                        |   <------   | 5.44e-3 |   ------>    |
> |                   | One-Per-Env                              | 4.20e-2 | 5.52e-3 | 1.90e-3|
> |                   | Pretrained-$f$-plus-trained-$g_e$     | **2.29e-3** | **1.45e-3** | **1.27e-3**|
> |NS              | Pretrained-$f$-only                        |   <------   |1.75e-1|    ------>  |
> |                   | One-Per-Env                              | 6.76e-2 | 1.70e-2 | 1.18e-3|
> |                   | Pretrained-$f$-plus-trained-$g_e$     | **1.37e-2** | **8.07e-3** | **7.14e-3**|
>
>
> 3. Additional remarks:
>
> - **Scale for the MSE error maps in Fig. 2**
>
>   The scale of GS error maps is [0, 0.6] (dark->bright), and the scale for NS is [0, 0.2]. We adapted the scale of each experiment to better intensify the visual contrast. We will add these details when possible.
>
> - **Not considering noisy data.**
>
>   The generalization problem addressed here corresponds to better exploiting the knowledge in known environments and extrapolating these knowledge to new environments with similar but different characteristics. Deterministic dynamics of this type are used in the majority of the models used to represent physical phenomena (in mechanics, fluid dynamics, geophysics, etc). This then already covers a very large set of situations. As mentioned in the limitation section, this is the setting used in the vast majority of ML analysis and modeling of dynamical systems - see the references line 414. The limitations mentioned in the discussion addressed the difference of complexity between dealing with simulations and real world data (we will make this clearer in the revised version). However, as also mentioned in this section, this is clearly beyond the scope of this paper.

---

> > ### Comment · Reviewer_juRp · 2021-08-23
> > **reply to response**
> >
> > **- On the originality of our results w.r.t. ref [4].**
> > I'm not an expert on the material in [4] so I defer to the authors here. I would recommend that this distinction (additive rather than compositional, and temporal-spatial regression, rather than classification) is highlighted in the main text.
> >
> > **- Proof of proposition 2.**
> > Thank you for proposing to update.
> >
> > **- Why the linear case motivates the use of eq. (7) for NNs.**
> > Sorry, your explanation here hasn't helped me understand precisely why you take equation (7). I was looking for something precise e.g. "starting with proposition 4 and applying blah, together with proposition 5 and substituting X, we get something of the form of equation (7)".
> >
> > I think this is actually a crucial point in the paper, because if I understand correctly, all experiments rely on (7), but I don't understand the link from the theory to equation (7). If this is not clarified, the paper is ultimately disjointed.
> >
> > **- Usefulness of theorems 2-4.**
> > I still feel a general reader won't benefit much from reading these theorems in the main text. Does a general reader need to know the exact expressions that appear in these statements? Would it not be easier to read if their results were summarized in prose form, perhaps with a single simple equation, with the details left to supplementary material?
> >
> > **- Significance of the results.**
> > > If one wants to summarize, throughout the experiments, the test error of LEADS no min is x1.2-x2.6 larger than LEADS, the test error of FT-NODE is x1.8-x4.9 larger than LEADS.
> >
> > I grudgingly accept in summary it seems better, though I still maintain that FT-NODE on GS test error of 3.86+-3.36e-3 is not significantly higher than LEADs, 2.08+-2.88e-3. I would want authors to at least comment on this specific result, even if their overall conclusion is that LEADs performs better in summary.
> >
> > **- Significance of the results compared to NN baselines.**
> > I agree that the comparisons are fair for the baselines considered. But my point was that in the small data regime, perhaps some other methods would be better suited than NNs. Adding a note on this would be sufficient for me on this point (no extra experiments required).
> >
> > **- Performance differences in different regimes w.r.t the number of environments and trajectories.**
> > It would be a stronger paper to have figure 4 with GS and NS in addition to LV. I recommend running this to confirm and adding to appendix, and noting result in main text.
> >
> > **- Results on GS and NS for Fig 5.**
> > Thanks for the extra results. I spotted that one-per-env is lower than LEADs at 10,000 steps for NS - why is that? Or is it perhaps a typo? If a typo, then I am happy with these results.
> >
> > **- additional remarks **
> > noted, thanks for the info.
> >
> > **Overall **
> > I would increase my score if I can get a precise explanation on where equation (7) comes from, and exactly how it relates to section 3.2

---

> > > ### Author Response · Authors · 2021-08-26
> > > **Response to the reply**
> > >
> > > Thank you for your comprehensive reply to our response, we appreciate your efforts and the time you have spent evaluating this work. We will make our best to clarify your points in detail below.
> > >
> > > The main point concerns the explanation for Eq. (7). This is addressed below and we answer the other points afterward.
> > >
> > > **A.**  We propose to explain how we reached Eq. (7) through a synthesis of the main results underlying our method and leading to this Eq. (7).
> > >
> > > We proceed in three steps: 1. A general result on complexity control with Proposition 2 and the inequality $\log\mathcal{C}\_{\hat{\mathcal{F}}}(\varepsilon, \hat{\mathcal{G}}) \leq \omega(r, \varepsilon)$ with $r=\sup\_{g\_e\in \hat{\mathcal{G}}}\Omega(g\_e)$ appearing after Line 189 for some abstract $\Omega$; 2. We illustrate our approach in the linear case with Proposition 4, proposing a simple $\Omega$ function in this context; 3. We instantiate $\Omega$ for Neural Networks in Eq. (7) – with Proposition S2 in Supplemental B.4.2.
> > >
> > > 1.   General result on complexity control
> > >
> > > Proposition 2 establishes a generalization bound when considering multiple, here $m$, environments. The main consequence of these results could be summarized as follows:
> > >
> > > - When we increase the number of environments $m$, the sample complexity will be reduced by a factor $1/m$ through the capacity of the shared function $f$ ($\mathcal{C}\_{\hat{\mathcal{G}}}(\varepsilon, \hat{\mathcal{F}})$), while the complexity for the environment specific functions $g\_e$ ($\mathcal{C}\_{\hat{\mathcal{F}}}(\varepsilon, \hat{\mathcal{G}})$), remains constant. While the shared component $f$ benefits from the multiple environment, the environment component $g\_e$ does not.
> > >
> > > - To control the bound, it is then relevant to control the complexity of functions $g\_e$, since it will not be reduced with the number of environments. This is formalized in the inequality after Line 189: $\log\mathcal{C}\_{\hat{\mathcal{F}}}(\varepsilon, \hat{\mathcal{G}}) \leq \omega(r, \varepsilon)$ with $r=\sup\_{g\_e\in \hat{\mathcal{G}}}\Omega(g\_e)$. Here $\Omega$ is an effective measure of the complexity of a function ($g\_e$ here), and $\omega$ is the capacity upper bound which decreases with $r$.
> > >
> > > This is the starting point of the idea developed further. We proceed as follows:
> > >
> > > - The difficulty in making use of the above inequality is to define an appropriate function $\omega(r, \varepsilon)$ which upper-bounds the $g\_e$ function capacity and which at the same time allows us to effectively control this capacity.
> > > - We show that for the linear case, it is possible to get a tight bound so that theory and practice agree.
> > > - For the nonlinear (NN) case, the bound is obtainable in the same way. Despite the fact that this bound has constants which may be large for the case of neural networks, we show experimentally that this bound leads to an effective complexity control and improves generalization. This is detailed below.
> > >
> > > 2. Linear case
> > >
> > > We show for this tractable case where the bounds are close to the actual capacity, that this idea can work (Figure 1). This validates, for this linear context, the ideas introduced in Proposition 2.
> > >
> > > - For Proposition 4, we exhibit here an $\omega(r, \varepsilon)$, which is a function of the measure of complexity $\Omega(g\_e)$. The latter writes as $\Omega(g\_e)=||\mathbf{G}||^2\_F$. By controlling the value $||\mathbf{G}||^2\_F\leq r$, it is then possible to control the complexity of the space where lies $g\_e$.
> > >
> > > - The linear case is detailed in the main text because 1) it gives an empirical validation of the general theoretical propositions and 2) following roughly the same derivation in the case of nonlinear (NN) functions, Eq. (7) is obtained. This linear case, thus provides a guide and an intuition for the more difficult nonlinear case. Note that Proposition 5 is a direct consequence of Proposition 4.
> > >
> > > 3. Instantiation for Neural Networks in Eq. (7)
> > >
> > > The derivation for the non linear case is independent of the linear instantiation, but the general idea is the same.
> > >
> > > - The main result is Proposition S2, Eq. (S3). Here we exhibit a bound  $\omega(R,L,\epsilon)$ which depends on two parameters $R$ and $L$. They will play the same role as the parameter $r$ for the linear case above. They respectively allow us to control the capacity through the infinity norm and the Lipschitz seminorm of $g_e$: $||g||_\infty\leq R$ and $||g||_\text{Lip}\leq L$.
> > > - These two norms appear naturally in the upper bound in Theorem S3, which is an adaptation of Theorem 11 in [S4], also referenced in the main text. The derivation appears in Section B.4.2 of Supplemental with the main results in Eq. (S4) and (S5). To summarize, one shows here that $\mathcal{C}_{\hat{\mathcal{F}}}(\varepsilon, \hat{\mathcal{G}})\leq c\_1 \log \frac{RL}{\varepsilon} + c\_2 =: \omega(R, L, \varepsilon)$ with two constants $c\_1, c\_2$ (details in Eq. (S4) and (S5)).
> > > - To optimize the latter bound, we then propose to consider $\Omega(g\_e)=||g\_e||\_\infty^2+\alpha||g\_e||\_{\text{Lip}}^2$. This is further refined in Proposition S3, which allows us to relate this derivation with equation $\log\mathcal{C}\_{\hat{\mathcal{F}}}(\varepsilon,\hat{\mathcal{G}}) \leq \omega(r, \varepsilon)$ (Line 189), thus providing a unified formulation for the linear and nonlinear cases.
> > >
> > > **Note on the paper presentation.** We agree that the transition from the formal results to the equation could have been better introduced. We will make the transition less abrupt by adding an explanatory paragraph at the beginning of Section 3.3.
> > >
> > > We propose to replace the first phrase by the following paragraph:
> > >
> > > “The above linear case validates the ideas introduced in Prop. 2 and provides an instantiation guide and an intuition on the more complex nonlinear case. This motivates us to instantiate the general case by choosing an appropriate approximating space $\hat{\mathcal{F}}$ and a penalization function $\Omega$ from generalization bounds for the corresponding space. Section B.4 of the Appendix contains additional details justifying those choices.”
> > >
> > > **B.** Other points
> > >
> > > - > **On the originality of our results w.r.t. ref [4]**. I'm not an expert on the material in [4] so I defer to the authors here. I would recommend that this distinction (additive rather than compositional, and temporal-spatial regression, rather than classification) is highlighted in the main text.
> > >
> > >   This could definitely clarify and highlight the contributions of our framework w.r.t. [4]; we will update our submission accordingly.
> > >
> > > - > **Usefulness of theorems 2-4**. I still feel a general reader won't benefit much from reading these theorems in the main text. Does a general reader need to know the exact expressions that appear in these statements? Would it not be easier to read if their results were summarized in prose form, perhaps with a single simple equation, with the details left to supplementary material?
> > >
> > >   We agree that practitioners may not be particularly interested with the exact quantitative bounds and derivations. However, we believe that others interested in continuing the theoretical work further will find these results particularly useful. What we could do is to provide a reading guide, in the same spirit as what we introduced above.
> > >
> > > - > **Significance of the results.**
> > > > > If one wants to summarize, throughout the experiments, the test error of LEADS no min is x1.2-x2.6 larger than LEADS, the test error of FT-NODE is x1.8-x4.9 larger than LEADS.
> > > >
> > > > I grudgingly accept in summary it seems better, though I still maintain that FT-NODE on GS test error of 3.86+-3.36e-3 is not significantly higher than LEADs, 2.08+-2.88e-3. I would want authors to at least comment on this specific result, even if their overall conclusion is that LEADs performs better in summary.
> > >
> > >   We propose to add the following sentence:
> > >   Note however that the variance computed over the trajectories makes some of the results not significantly different, e.g. FT-NODE vs LEADS on the GS example.
> > >
> > > - > **Significance of the results compared to NN baselines**. I agree that the comparisons are fair for the baselines considered. But my point was that in the small data regime, perhaps some other methods would be better suited than NNs. Adding a note on this would be sufficient for me on this point (no extra experiments required).
> > >
> > >   Here is a possible note highlighting this point. The theory developed here is quite general and not limited to NNs. We chose the latter since they represent a flexible family of hypotheses, and their complexity can be efficiently controlled in different ways, as we showed in our experiments. However, it is possible that less expressive families might do as well or better in low data regimes.
> > >
> > > - > **Performance differences in different regimes w.r.t the number of environments and trajectories**. It would be a stronger paper to have figure 4 with GS and NS in addition to LV. I recommend running this to confirm and adding to appendix, and noting result in main text.
> > >
> > >   This is extremely heavy in terms of computations. We will try to confirm on GS and NS, but this will take time and this will not be available we think in the next few days.
> > >
> > > - > **Results on GS and NS for Fig 5.** Thanks for the extra results. I spotted that one-per-env is lower than LEADs at 10,000 steps for NS - why is that? Or is it perhaps a typo? If a typo, then I am happy with these results.
> > >
> > >   Thank you for pointing this out. It is actually a typo, the test loss of One-Per-Env. for NS at the 10000-th step is 1.18e-2.

---

> > > > ### Comment · Reviewer_juRp · 2021-09-02
> > > > **increasing my score**
> > > >
> > > > Thanks for the detailed reply, and for taking my suggestions into account. Sorry to keep you waiting on my response.
> > > >
> > > > I now understand more where (7) comes from, which was my main confusion. Your proposal for replacing the first phrase above (7) looks good to me, and I think adding the reading guide will really help many readers.
> > > >
> > > > Do not worry about the GS and NS results in time for this discussion period, but I would encourage you to run this by camera-ready time.
> > > >
> > > > Based on the changes you intend to make, I have now increased my score from 4 to 7.

---

> > > > > ### Author Response · Authors · 2021-09-02
> > > > > **Thanks for the fast answer**
> > > > >
> > > > > Thanks, and thanks a lot again for your time and consideration

---

> ### Author Response · Authors · 2021-09-02
> **response to additional questions**
>
> Dear reviewer,
>
> In our 26/08 response, we tried to answer precisely your central question and your other comments.
> Since the discussion period ends today, 2/09, please let us know if we have clarified your points or if you would like complementary information.
>
> Thanks again for your time.

---

### Author Response · Authors · 2021-08-10
**To all reviewers**

First of all, we thank all the reviewers for their careful reading of the manuscript and their constructive remarks. We provide detailed answers in the comments section of each review.

We have also added in the comments new experimental results as suggested by the reviewers. They concern:
- Results on learning dynamics in new environments for GS and NS datasets. (See the response to Reviewer juRp)
- Results with different numbers of environments and trajectories in Figure 4 for FT-RNN and FT-NODE baselines. (See the response to Reviewer 27Ns).

One may also find some substantial changes that have been made w.r.t the past submission in the Submission History.

---

### Decision · Program_Chairs · 2021-09-27

**Decision:**

Accept (Poster)

**Comment:**

From the SAC: This is an instance where a good rebuttal has helped. The original decision on this paper was a reject, but the SAC felt that your rebuttal was very thorough and detailed. Hence I am recommending moving this paper to an accept. Please do make sure in the next version of the paper to take all reviewer comments into account as well as to merge into the paper the additional discussion in your rebuttal (perhaps, due to space considerations, into the appendices).